# Discovery of superconductivity in quasicrystal

K. Kamiya[1,5], T. Takeuchi[2], N. Kabeya[3], N. Wada[1], T. Ishimasa[4], A. Ochiai[3], K. Deguchi[1], K. Imura[1] & N.K. Sato[1]

Superconductivity is ubiquitous as evidenced by the observation in many crystals including carrier-doped oxides and diamond. Amorphous solids are no exception. However, it remains to be discovered in quasicrystals, in which atoms are ordered over long distances but not in a periodically repeating arrangement. Here we report electrical resistivity, magnetization, and specific-heat measurements of Al–Zn–Mg quasicrystal, presenting convincing evidence for the emergence of bulk superconductivity at a very low transition temperature of $T_c \cong 0.05$ K. We also find superconductivity in its approximant crystals, structures that are periodic, but that are very similar to quasicrystals. These observations demonstrate that the effective interaction between electrons remains attractive under variation of the atomic arrangement from periodic to quasiperiodic one. The discovery of the superconducting quasicrystal, in which the fractal geometry interplays with superconductivity, opens the door to a new type of superconductivity, fractal superconductivity.

[1] Department of Physics, Graduate School of Science, Nagoya University, Nagoya 464-8602, Japan. [2] Toyota Technological Institute, Nagoya 468-8511, Japan. [3] Department of Physics, Graduate School of Science, Tohoku University, Sendai 980-8578, Japan. [4] Toyota Physical and Chemical Research Institute, Aichi 480-1192, Japan. [5] Present address: UACJ Corporation, Nagoya 455-8670, Japan. Correspondence and requests for materials should be addressed to N.K.S. (email: kensho@cc.nagoya-u.ac.jp)

In classical crystallography, a crystal was defined as a periodic arrangement of atoms with translational periodicity, leading to an infinitely extended crystal structure by aligning building blocks called unit cells; as an example, we illustrate a cubic unit cell (Fig. 1a), in which the corner and body-centered positions are occupied by the icosahedron. This traditional definition was forced to modify by the discovery of quasicrystal (QC) by Shechtman et al.[1], which led to a paradigm shift in science. Nowadays, QC is understood as a structure that is long-range ordered (as manifested in the occurrence of sharp diffraction spots) but not periodic (Fig. 1b)[2–4]. Another characteristic of QC is the presence of a non-crystallographic rotational symmetry[2,3]; whereas periodic crystals can possess only two-, three-, four-, and sixfold rotational symmetries, icosahedral QCs have fivefold symmetry (Fig. 1b). In recent years, cold atom gaseous QCs are formed in quasiperiodic optical potentials[5,6].

For simplicity, we consider a one-dimensional (1D) analog to QC known as the Fibonacci chain, $LSLLSLSLLSLLS...$ (see QC in Fig. 1c)[7], where $L$ and $S$ are long and short segments with the ratio $L/S$ equal to the golden mean $\tau \equiv (1 + \sqrt{5})/2$. This chain looks to have no order at a glance, but it has the perfect order as understood from the fact that it was created by successively applying the self-generation rule, $L \rightarrow LS$ and $S \rightarrow L$, onto the first generation sequence, $L$, as demonstrated below,

$$L \ (\text{1st}) \rightarrow LS \ (\text{2nd})$$
$$\rightarrow LSL \ (\text{3rd}) \rightarrow LSLLS \ (\text{4th}) \rightarrow LSLLSLSL \ (\text{5th}) \rightarrow \cdots.$$

It may be noticed that $n$-th generation sequence is produced by placing $(n-2)$-th one on the right-hand side of $(n-1)$-th one. Then, the total number of the $L$ and $S$ segments of the $n$-th generation, $F_n$, follows the relation,

$$F_n = F_{n-1} + F_{n-2} \ (n \geq 3). \tag{1}$$

This recurrence relation with $F_1 = 1$ and $F_2 = 1$ gives the Fibonacci sequence, 1, 1, 2, 3, 5, 8, ... A series of the successive Fibonacci number ratio, $F_{n-1}/F_{n-2}$, approximates the golden ratio; 1/1, 2/1, 3/2, 5/3,..., $\lim_{n \to \infty} F_{n-1}/F_{n-2} = \tau (= 1.6180...)$. There is an actual material that corresponds to each rational ratio and is called approximant crystals (ACs). Examples are shown in Fig. 1c; 1/1AC is a periodic crystal consisting of the unit cell $LS$, 2/1AC consisting of $LSL$, and so on. In $F_{n-1}/F_{n-2}$ AC, $F_{n-1}$ and $F_{n-2}$ indicate the number of $L$ and $S$ segments contained in the unit cell, respectively. (In the 3D case, for example, 1/1AC denotes cubic 1/1-1/1-1/1AC.) This means that the unit cell size of AC increases with the order of the rational approximant.

Reflecting such the unique geometry, QC is expected to have an electronic state called critical state that is neither extended nor localized. The existence of extended eigenstates in periodic crystals is a consequence of Bloch's theorem, whereas in random

systems, strong disorder can lead to the formation of localized eigenstates, i.e., Anderson localization, which occurs due to the interference effect between propagating and backwards scattered waves. In QCs, critical eigenstates emerge as a result of the competition between the broken translational invariance and the self-similarity of quasiperiodic structure[8]. Besides extensive studies, the electronic state of QCs is veiled in mystery[9]. For example, an electronic long-range-ordered states is not established yet although it was observed in ACs[10,11]; to the best of our knowledge, there is no QC presenting the convincing evidence for bulk superconductivity[12–14], i.e., zero resistivity, Meissner effect, heat capacity jump, and the fivefold rotational symmetry as well. (In ref. [14], $Mg_3Zn_3Al_2$ was considered as a superconducting QC, but it seems to be AC according to the phase diagram given in ref. [15] and the present study, see below.) It is therefore interesting to discover superconductivity in QC. It is also interesting to examine whether the emerging superconductivity shows weak-coupling, spatially extended Cooper pairs or strong-coupling, local pairs (reflecting the critical state).

Here, we study the Al–Zn–Mg system as a test material owing to two reasons: First, it contains both QC[15,16] and AC phases[15,17,18], and, second, the AC phase exhibits superconductivity[14]. We show that bulk superconductivity emerges at $T_c \cong 0.05$ K in the Al-Zn-Mg QC, implying that it is not only the first superconducting QC but also the first QC exhibiting the electronic long-range order. We also show that temperature dependences of the thermodynamic properties and the upper critical filed are understood within the weak-coupling framework of superconductivity, suggesting the formation of spatially extended pairs.

## Results

**Sample characterization.** Samples prepared here are Al–Zn–Mg-based QC, 2/1AC, and 1/1ACs, which are summarized in the ternary phase diagram (Fig. 2). As reported in ref. [15], the 1/1ACs have a wide composition range. In this paper, each 1/1AC sample with different composition is identified using the alphabetical character, e.g., 1/1AC_A. The 1/1AC_G is a mother alloy of the QC and has almost the same composition as the 2/1AC. Note that the alloy $Mg_3Zn_3Al_2$ mentioned above is close to the 1/1AC_E sample.

The structure of the obtained samples was studied by X-ray and electron diffraction method. The lattice constant $a$ of the 1/1AC samples is illustrated in Fig. 3a as a function of Al content. We note that $a$ decreases almost linearly with the Al content.

For the QC, the following indexing scheme of the reflection vector $\mathbf{g}$ is used in this paper;

$$\mathbf{g} = \frac{1}{a_{6D}} \sum_{i=1}^{6} m_i \mathbf{e}_{i\parallel}. \tag{2}$$

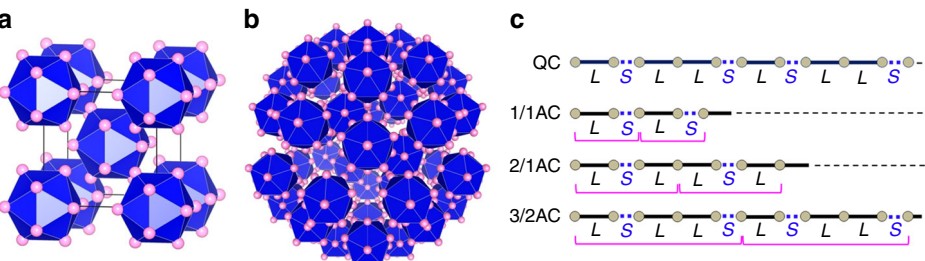

**Fig. 1** Periodic and quasiperiodic arrangement of atoms. **a** An example of cubic unit cell in which the icosahedron occupies the corner and body-centered positions. Pink balls indicate atoms. **b** An example of the Tsai-type icosahedral quasicrystal. The fivefold rotational symmetry and the self-similarity may be observed. **c** Fibonacci sequence and its approximants. Atoms are denoted by circles, and the long and short interatomic segment is denoted by $L$ and $S$, respectively. The bracket (e.g., $LS$ for 1/1AC and $LSL$ for 2/1AC) denotes the unit cell

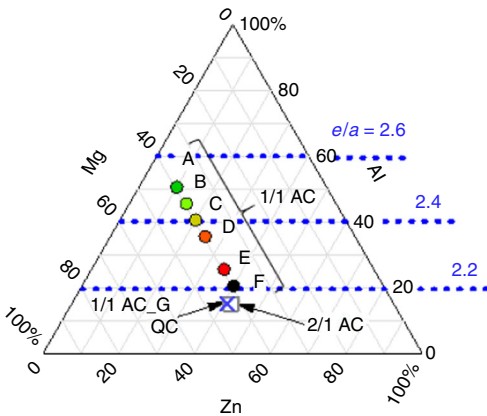

**Fig. 2** Ternary phase diagram. The samples studied here are summarized in the ternary phase diagram. The 1/1AC samples have a wide composition range: Each 1/1AC sample with different composition is identified using the alphabetical character. The composition of both the QC and the 1/1AC_G (the mother alloy of the QC) is $Al_{14.9}Mg_{44.1}Zn_{41.0}$, and that of the 2/1AC is $Al_{14.9}Mg_{43.0}Zn_{42.1}$. The ratio $e/a$ denotes electron number per atom

Here, the set of integers, $m_i$, represents reflection index. The vectors $\mathbf{e}_{i\parallel}$ have a length equal to $1/\sqrt{2}$, and they are parallel to the lines connecting the center of an icosahedron and the surrounding six vertices as in Fig. 6 of ref. [19]. The lattice parameter $a_{6D}$ of the 6D hypercubic lattice may be related to the edge length $a_R$ of the rhombohedral cells of the 3D Penrose tiling as follows,

$$a_R = a_{6D}/\sqrt{2}. \qquad (3)$$

In the following, we focus on the QC, the 2/1AC, and the 1/1AC_G (the mother alloy of the QC). The representative X-ray diffraction patterns of them are displayed in Fig. 3b, confirming almost the single phase. The diffraction peaks of the QC were indexed using the 6D lattice parameter, $a_{6D} = 0.7308 \pm 0.0001$ nm. Absence of any extinction condition indicates P-type of icosahedral QC. For the 2/1AC, the peaks are labeled using the lattice parameter, $a_{2/1} = 2.3006 \pm 0.0004$ nm, indicating P-type cubic phase. The intensity is strong for the 850 reflection ($d = 0.244$ nm) and the 583 reflection ($d = 0.232$ nm). Note that these indices are combination of successive Fibonacci numbers. For the 1/1AC_G, similar results are obtained: The peaks were indexed with the lattice parameter, $a_{1/1} = 1.4195 \pm 0.0003$ nm, for I-type cubic crystal. Note that the 530 reflection with the spacing $d = 0.243$ nm and the 352 reflection with $d = 0.230$ nm have a strong intensity. These indices are combination of the successive Fibonacci numbers again, and they correspond to 211111 and 221001 reflections of the QC, respectively.

The unit cell sizes obtained above satisfy the following equation [20,21],

$$a_{F_{n-1}/F_{n-2}} = \sqrt{2/(2+\tau)}(F_{n-1}\tau + F_{n-2})a_{6D}. \qquad (4)$$

This ensures our assignment of the QC and the AC samples.

Using the X-ray diffraction peaks around $2\theta = 65°$, we estimated the correlation length as 47, 85, and 28 nm for the QC, the 2/1AC, and the 1/1AC_G, respectively. Comparison between them suggests that the 1/1AC_G is meta-stable at the composition $Al_{14.9}Mg_{44.1}Zn_{41.0}$. For further comparison, we evaluated the correlation length of the 1/1AC_A as more than 65 nm, which is twice the 1/1AC_G value. This difference in the

sample quality would yield the sample dependence in the physical properties among the different ACs (Supplementary Figure 2).

Electron diffraction patterns of the QC, the 2/1AC, and the 1/1AC_G are demonstrated in Fig. 3c–g. Figure 3c displays a fivefold diffraction pattern of symmetry $m\overline{35}$ of the QC. Indices of reflections A and B are $1220\overline{1}0$ and 221001, respectively. Magnified image (Fig. 3d) including reflection B shows deviation from the exact regular pentagon for weaker reflections, indicating the presence of linear phason strain [22]. Figure 3e shows a twofold diffraction pattern of the QC. Indices of reflections A and C are $1220\overline{1}0$ and $121\overline{1}\overline{1}1$, respectively. The $\tau^3$-scaling agrees with P-type of icosahedral QC. Figure 3f and g shows diffraction patterns of the 2/1AC and the 1/1AC_G, respectively, with the incident beam along each [001] direction. The 2/1 and 1/1ACs show no fourfold but twofold axis. In Fig. 3f, indices of reflections D and E are 10 00 and 850 of 2/1AC, respectively. We observe the reflection condition that $h$ is even for $h00$ and $hk0$ reflections. The $0k0$ reflections with odd $k$ should disappear following this reflection condition, but they are actually observed due to multiple diffraction effects. This observation is consistent with the space group $Pa\overline{3}$ proposed for the 2/1AC [17]. In Fig. 3g, reflections F and G correspond to 600 and 530 reflections of the 1/1AC, respectively. Note reflection condition of $h + k + l =$ even for $hkl$ reflection, which is consistent with the reported space group $Im\overline{3}$ [18].

**Electrical resistivity.** Figure 4a shows the electrical resistivity normalized by the resistivity at $T = 280$ K, $\rho/\rho_{280\,K}$, as a function of temperature $T$ in a logarithmic scale. Three points are to be noted. First, all the materials studied here show zero resistivity. Second, $\rho_{280\,K}$ of the QC and the 2/1AC amounts to ~150 μΩ cm, greater than that of all the 1/1AC samples (inset of Fig. 4a). Third, while all the 1/1AC samples present the metallic behavior, the QC and the 2/1AC show the negative temperature coefficient of resistivity, $d\rho/dT < 0$ (Fig. 4b).

The normal state conductivity of QCs has been sometimes discussed using the concept of the Anderson localization [9]. For the present case, it remains open if the second and third points mentioned above show a precursor of the electron localization in the QC and the 2/1AC. This should be examined in the future by virtue of phason-strain-free samples; the present sample contains a linear phason strain as mentioned above.

**Specific heat in normal state.** The temperature dependences of the specific heat $C(T)$ in the normal state of the QC, the 2/1AC, and the representative 1/1AC samples are shown in Fig. 5a in the form of $C/T$ vs $T^2$. Using the relation, $\frac{C}{T} = \gamma + \beta T^2$, we obtain the coefficients $\gamma$ and $\beta$ for each sample. The Debye temperature $\Theta_D$ is deduced from $\beta$ and plotted in Fig. 5b as a function of Al content, in good agreement with the previous report [15]. We confirm that $\Theta_D$ is almost independent of Al content. For the Al-content dependence of the electronic-specific heat coefficient $\gamma$, see below.

**Relation between $T_c$ and $1/\gamma$.** Figure 6a shows the superconducting transition temperature $T_c$ defined by zero resistivity as a function of Al content. Note that the zero resistivity corresponds to the heat capacity jump (see below) and hence shows the bulk transition of superconductivity. As Al content is decreased, $T_c$ is monotonically decreased from ~0.8 to ~0.2 K, followed by the sudden drop down to ~0.05 K at 15% Al content (corresponding to the QC, the 2/1AC, and the 1/1AC_G).

Figure 6b shows the Al-content dependence of the electronicspecific heat coefficient $\gamma$ deduced from Fig. 5a. We observe that $\gamma$ monotonically decreases with Al content, suggesting that

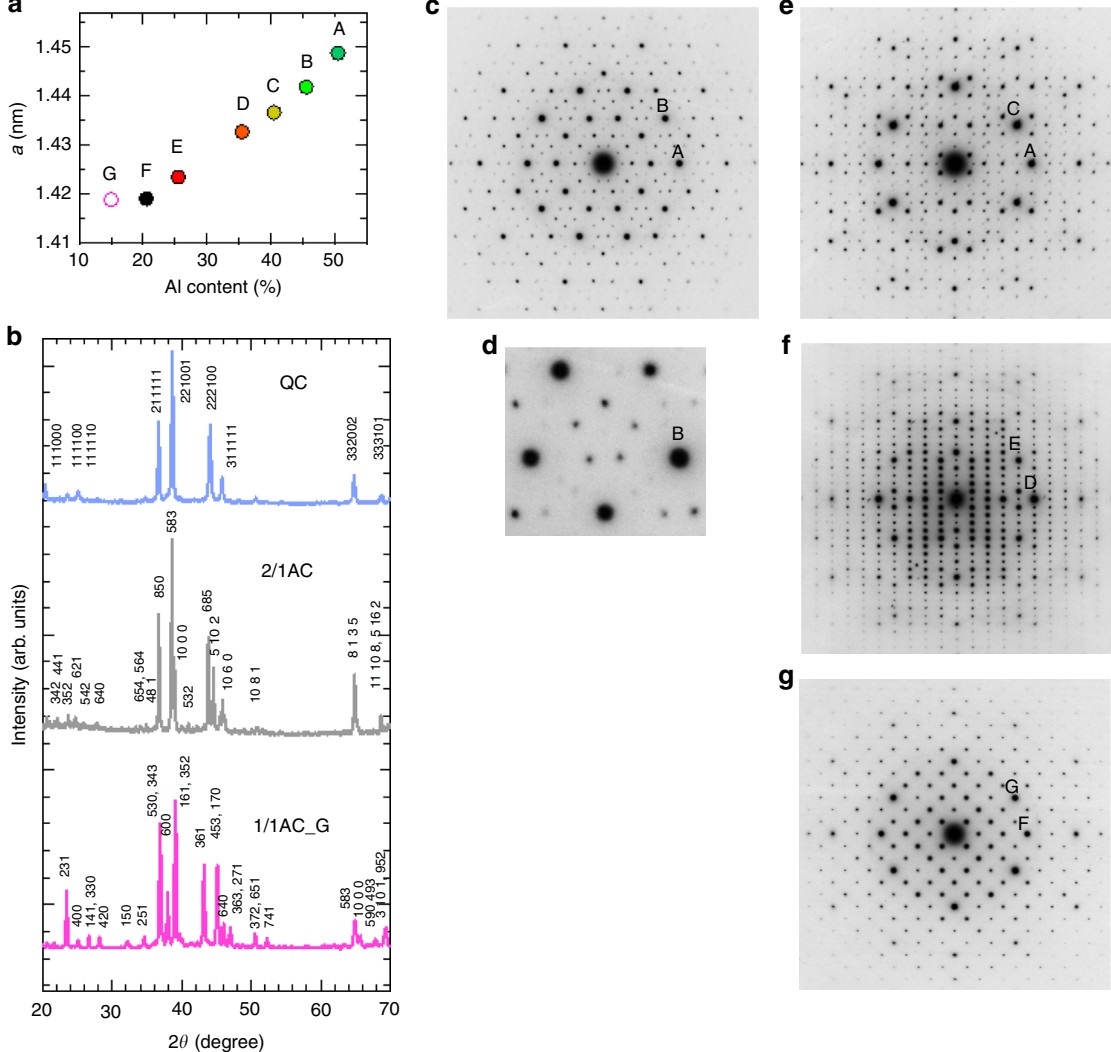

**Fig. 3** Diffraction patterns. **a** Lattice parameter $a$ of 1/1AC samples determined from X-ray diffraction method as a function of Al content. Note that $a$ decreases almost linearly with Al content. **b** Representative X-ray diffraction patterns of the QC, the 2/1AC, and the 1/1AC_G. Note that six integers are needed to index the peak of the QC. **c**, **d** Fivefold electron diffraction patterns of the QC. Magnified image (**d**) shows deviation from the perfect regular pentagon for weaker reflections. **e** Twofold electron diffraction pattern of the QC. **f**, **g** Electron diffraction patterns of 2/1 and 1/1AC with the incident beam along each [001] direction

the density of states at the Fermi energy $E_F$, $D(E_F)$, decreases with Al content. Note that $\gamma$ slightly drops at 15% Al content, which is likely related to the electronic stabilization effect, i.e., the pseudogap formation due to the so-called Hume–Rothery mechanism[15].

To see the relation between $T_c$ and $\gamma$, we plot $\ln T_c$ vs $1/\gamma$ in Fig. 6c with Al content as an implicit parameter. We find that all the samples lie on the straight line within an experimental uncertainty. According to the BCS theory, $T_c$ is given as follows,

$$T_c = 1.14\Theta_D e^{-1/VD(E_F)}. \tag{5}$$

Here, $V$ is the effective electron–electron interaction with the weak-coupling condition $|VD(E_F)| \ll 1$. As $\Theta_D$ is almost independent of Al content in the present system as mentioned above, Eq. (5) leads to the relationship, $\ln T_c \propto 1/\gamma$, if $V$ is the same among the samples. This is just observed here, meaning that the effective interaction $V$ remains attractive and unchanged in magnitude under variation of the atomic arrangement from the AC to the QC and $T_c$ is fully determined by $D(E_F)$.

**Bulk transition of superconductivity in QC.** Let us focus on the superconducting transition of the QC. (See Supplementary Figure 2 for the AC samples.) At $T_c$ marked by the resistivity drop (Fig. 7a), the real part of the ac magnetic susceptibility ($\chi'$) becomes negative (Fig. 7b), signaling the shielding effect associated with the zero resistivity. Upon cooling the sample through $T_c$ under an external magnetic field, the dc magnetization $M$ becomes diamagnetic (Fig. 7c), indicating the exclusion of the magnetic flux due to the Meissner effect. As seen in Fig. 7d, the specific heat divided by temperature $C_e/T$ shows the large jump ($\delta C_e/T_c \sim 1.2\gamma$) at $T_c$, where $C_e$ denotes the electronic part of specific heat, obtained by subtracting the lattice contribution from the measured specific heat, and $\delta C_e$ indicates the jump height of $C_e$. This indicates that almost all mobile electrons in the sample participate in the superconductivity. These provide convincing evidence for the emergence of bulk superconductivity in the QC.

In Fig. 8, we show the normalized specific heat $C_e/\gamma T$ of the QC and the 1/1AC_A as a function of the reduced temperature $t = T/T_c$. (The QC sample presented here is different from that shown in Fig. 7d.) We observe that the data of the QC and the 1/1AC are in good agreement with each other. Note that both the

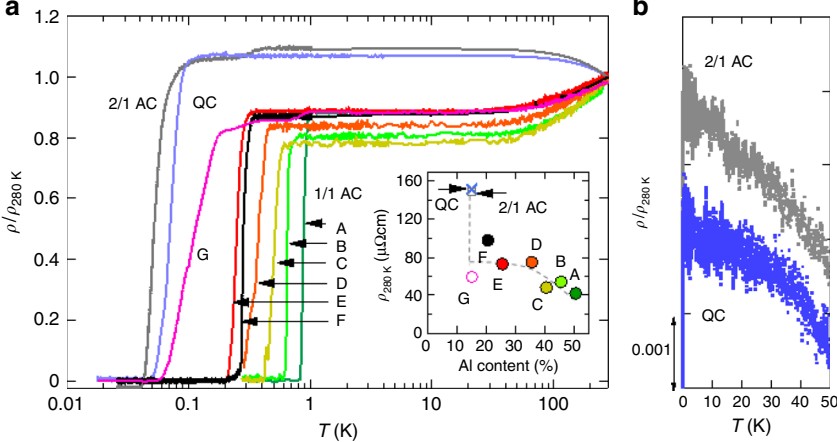

**Fig. 4** Electrical resistivity. **a** Temperature dependence of normalized electrical resistivity. $\rho_{280\,K}$ denotes the resistivity at $T = 280$ K. The samples plotted are the as-cast 1/1AC_A, E, F, G samples, the annealed 1/1AC_B, C, D samples, the 2/1AC sample, and the QC sample. The 1/1AC_G is a mother ingot of the QC. Inset: Electrical resistivity at $T = 280$ K as a function of Al content. The broken line is a guide to the eyes: note that the resistivity of the QC and the 2/1AC is larger than that of the 1/1AC samples. **b** Temperature dependence of normalized resistivity of the QC and the 2/1AC in an expanded scale below 50 K. The data are shifted vertically for clarity. Note that they show the negative temperature coefficient of resistivity

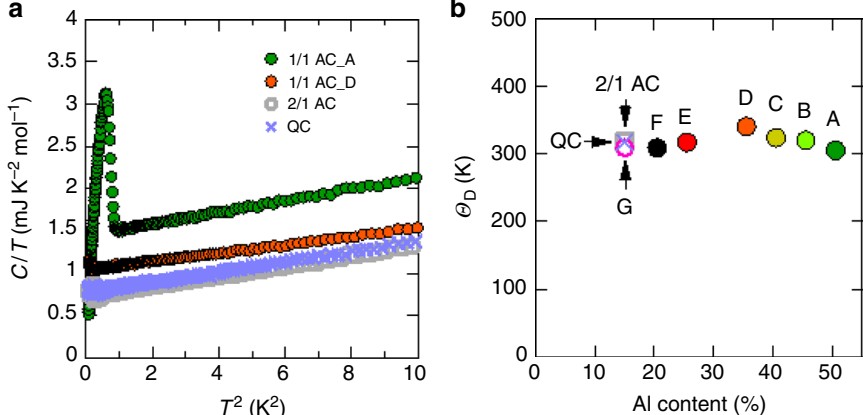

**Fig. 5** Specific heat in normal state. **a** Temperature dependences of specific heat in the form of $C/T$ vs $T^2$ in the range of $0 < T^2 < 10$. The results obtained here are consistent with the data in ref. [15]. Note that all the samples show a conventional form of $\frac{C}{T} = \gamma + \beta T^2$ in the normal state. The peak anomaly at low temperature is due to the onset of superconductivity. **b** Debye temperature $\Theta_D$ deduced from $\beta$. Note that $\Theta_D$ is almost independent of Al content

results are compatible with the BCS theory (see solid line), the only available theory at present for comparison with the experiment, although the base temperature of the experiment is not low enough to confirm the exponential tail of $C_e(t)$ at very low temperatures. The agreement with the theory signifies the onset of long-range order of Cooper pairs with opening of a full gap $\Delta$ characterized by the relation $2\Delta = 3.5k_B T_c$ (where $k_B$ is the Boltzmann constant).

**Superconducting critical field**. The magnetic field dependence of the electrical resistivity $\rho(H)$ is demonstrated in Fig. 9. The zero resistivity defines the upper critical field $H_{c2}$ shown in the inset of Fig. 10. Note that the 1/1AC_F has a several times larger $H_{c2}$ than Al metal, while it has a several times lower $T_c$. This excludes the possibility that the superconductivity might arise from Al-derived impurity phase. Combining the relations, $\kappa = H_{c2}(0)/\sqrt{2}H_c(0)$ and $H_c(0) = T_c\sqrt{5.94\gamma}$ (where $\kappa$ is the so-called GL parameter, $H_{c2}(0)$ and $H_c(0)$ are the upper and the thermodynamic critical fields extrapolated to zero temperature, respectively), we evaluate $\kappa$ as 136, 128, and 337 for the QC, the 2/1AC, and the 1/1AC_F,

respectively. These values confirm that the present system is a type-II superconductor, in which the magnetic field penetrates the sample. The coherence length $\xi(0)$ was also evaluated from the relation $H_{c2}(0) = \phi_0/2\pi\xi(0)^2$ (where $\phi_0$ is the flux quantum) as $\xi(0) \sim 139$, 143, and 83 nm for the QC, the 2/1AC, and the 1/1AC_F, respectively.

The reduced upper critical field is defined as $h = -H_{c2}/(T_c dH_{c2}/dT|_{T=T_c})$, and is plotted in Fig. 10 as a function of the reduced temperature ($t = T/T_c$). We compare $h(t)$ with Werthamer–Helfand–Hohenberg (WHH) theory[23], which takes into account of electron mean free path ($l$), spin–orbit scattering, and spin paramagnetism. The experimental results are in good agreement with the theory (solid line) for the case of no spin paramagnetic or spin–orbit effects and in the dirty limit ($\xi(0) \gg l$), in which scattering from physical and chemical impurities is large compared with the superconducting energy gap. This dirty-limit superconductivity seems compatible with the large coherence length estimated above and the large residual resistivity (i.e., small mean path) shown in Fig. 4. On the other hand, the present system is distinguished from some dirty

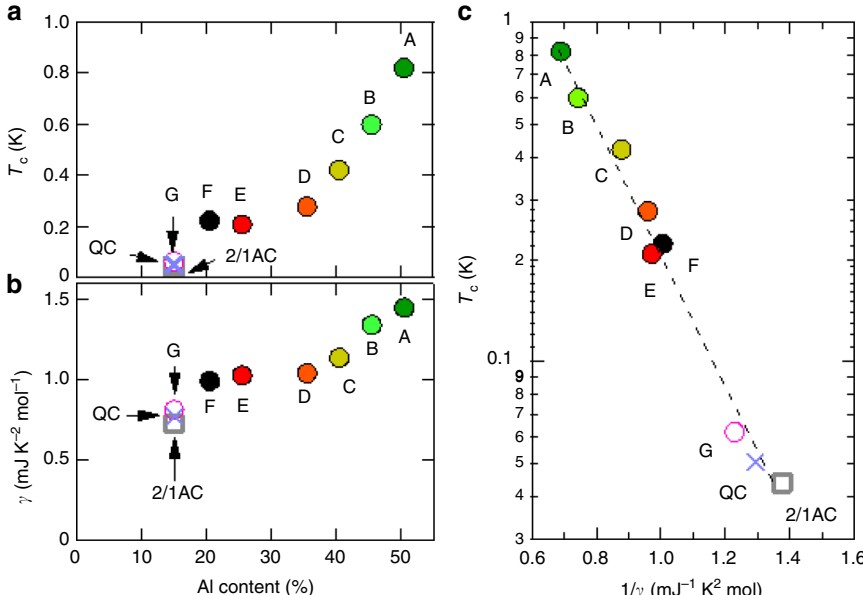

**Fig. 6** Relation between transition temperature and $\gamma$ coefficient. **a** Superconducting transition temperature $T_c$ as a function of Al content. Note that $T_c$ shows the sharp drop at Al content of about 15% corresponding to the QC and the 2/1AC. **b** Electronicspecific heat coefficient $\gamma$ as a function of Al content. The slight drop at 15% Al content suggests the pseudogap formation. **c** Correlation between $T_c$ and $1/\gamma$. The straight line denotes the interrelationship, $\ln T_c \propto 1/\gamma$, marking the constant pairing interaction

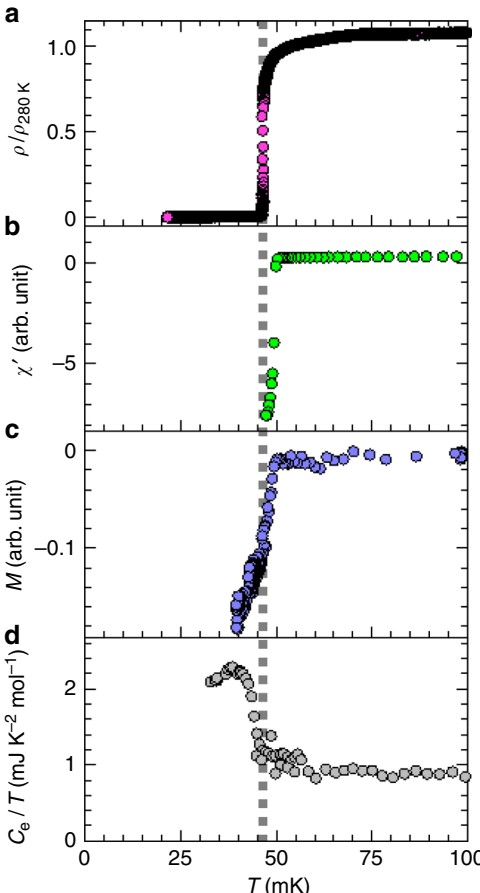

**Fig. 7** Physical properties around $T_C$ of quasicrystal. Temperature dependence of the normalized electrical resistivity (**a**), the real part of the ac magnetic susceptibility (**b**), the dc magnetization at external magnetic field of approximately 4 mOe (**c**), and the electronic part of the specific heat divided by temperature (**d**), for the QC sample. The broken line marks the transition temperature $T_c \approx 0.05$ K

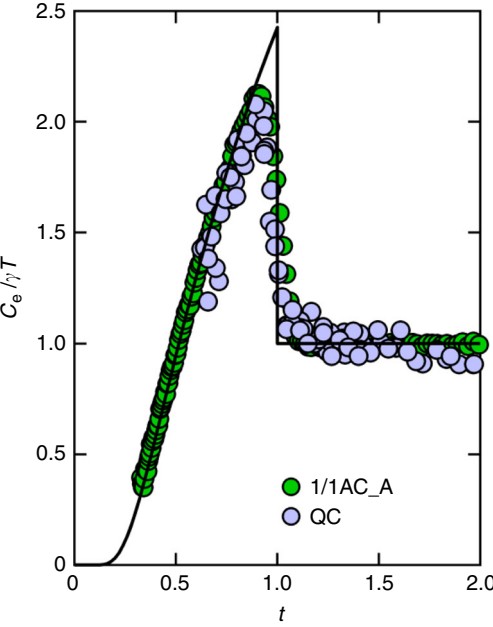

**Fig. 8** Specific heat around superconducting transition. Normalized specific heat divided by temperature as a function of the reduced temperature $t = T/T_c$ for the QC (with $T_c \approx 0.05$ K) and the 1/1AC_A ($T_c \approx 0.8$ K). Here, $C_e$ is the electronic part of the specific heat. The solid line denotes the weak-coupling BCS theory. Note that both results of the QC and 1/1AC samples are compatible with the weak-coupling theory

systems[24] in which $h(t)$ was enhanced over the WHH theory as a result of the field-induced suppression of localization.

## Discussion

In general, superconductivity needs the attractive interaction among electrons and the finite density of states at the Fermi

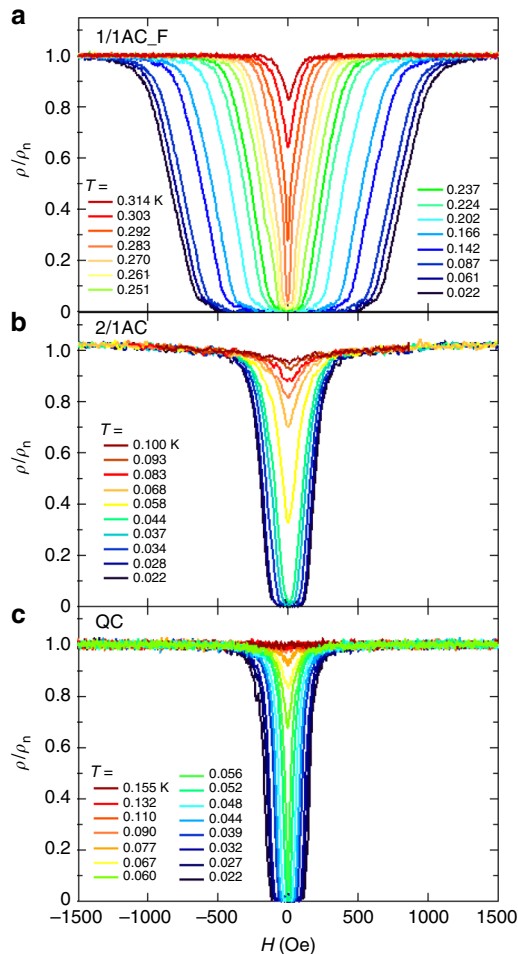

**Fig. 9** Magnetoresistance at a constant temperature. Magnetic field dependence of normalized electrical resistivity of the 1/1AC_F sample (**a**), the 2/1AC sample (**b**), and the QC sample (**c**). $\rho_n$ denotes the normal state resistivity

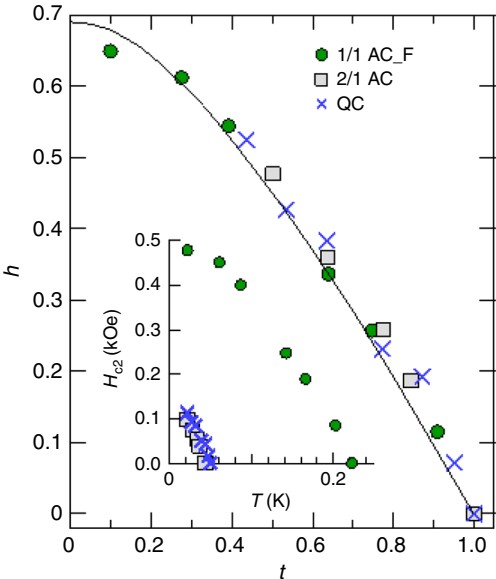

**Fig. 10** Superconducting upper critical field. The superconducting upper critical fields $H_{c2}$ of the QC, the 2/1AC, the 1/1AC_F are plotted in the form of $h = -H_{c2}/(T_c dH_{c2}/dT|_{T=T_c})$ vs $t = T/T_c$. The solid curve denotes the Werthamer–Helfand–Hohenberg theory in the dirty limit. The inset shows $H_{c2}(T)$ of the same samples as in the main frame. Note that the upper critical field in the limit of zero temperature of the 1/1AC_F ($H_{c2}(0) \cong 500$ Oe) substantially exceeds that of metal Al ($\sim 100$ Oe), excluding the possibility that the superconductivity would arise from Al-derived impurity phase

energy, i.e., $VD(E_F)>0$ in Eq. (5). For the present QC case, two points are to be noted. First, $D(E_F)$ is reduced presumably due to the pseudogap formation, and $T_c$ is much smaller than that of the ACs but remains finite. (The very low $T_c$ due to the pseudogap may explain why superconductivity was hardly observed in QCs.) This situation resembles that in superconductors in which charge-density-wave (CDW) states coexist; the Cooper pairing and the CDW instabilities compete for the Fermi surface and so the presence of the CDW depresses $T_c$.[25] Second, the fact that $V$ remains intact in the QC leads to the following discussion: The electron–electron interaction is expressed as $V = V_a - V_C$, where $V_a$ is the attractive pairing interaction (mediated by phonons in conventional BCS superconductors) and $V_C$ is the effective Coulomb repulsion. If the critical eigenstates of QCs would lead to the localization effect and, as a result, cause slow diffusion of electrons, then $V_C$ could be enhanced and $V$ would be reduced.[26,27] The absence of such the reduction in $V$ implies that the critical eigenstates would not have a dominant role in the superconductivity of the present QC.

In this study, we found no difference between the Al–Zn–Mg QC and other weak-coupling superconductors. According to a theoretical study by Sakai et al.[28], however, the Cooper pairs in the Penrose lattice are unconventional because the lack of the translational symmetry does not allow the conventional Cooper pairing formed at the opposite Fermi momenta, **k** and −**k**. It

would be challenging to detect the fractal superconducting order parameter as predicted by the theory. We hope that the present study stimulates a further work to reveal this new type of superconductivity.

## Methods

**Sample preparation.** The 1/1AC samples were prepared by induction melting of appropriate amounts of constituent elements, 99.99% Al, 99.9% Mg, and 99.99% Zn, in a boron-nitride crucible under Ar atmosphere[15]. Some of them were annealed at 300 °C for 6 h or at 360 °C for 5 h. The mother alloys of the QC ($Al_{14.9}Mg_{44.1}Zn_{41.0}$) and the 2/1AC ($Al_{14.9}Mg_{43.0}Zn_{42.1}$) were first prepared by induction melting of the constituent elements[15]. Then, by melt spinning of each mother alloy[15], the ribbon specimens were fabricated. Finally, the QC samples were obtained by sintering the ribbons at 300 °C and at 50 MPa for 1 h using a spark plasma sintering apparatus, whereas the 2/1AC samples were obtained by sintering the ribbons at the same conditions as the above and subsequently annealing the sintered ribbons at 300 °C for 5 h. Some of the QC samples were annealed at 360 °C for 5 h, whose structure was confirmed to be kept in the QC.

**Sample characterization.** The composition of the obtained samples was analyzed by using inductively coupled plasma (ICP) spectroscopy and scanning electron microscope (SEM). For the ICP, the analyzed composition agreed well with the nominal one within the error <2%, and no segregation was detected for the SEM within the experimental accuracy (Supplementary Figure 1).

Selected-area electron diffraction patterns were observed using a JEOL JEM-200CS microscope with a double tilting stage at the acceleration voltage 200 kV. The alloy specimens were crushed into fragments using an agate mortar and pestle, and transferred on a micro-grid mesh for the electron microscopic observation.

X-ray diffraction patterns were obtained using a RIGAKU IIB diffractometer. Lattice parameters of the QC and the ACs were determined from angles of Bragg reflections, $\theta$, using the extrapolation method: Least square fitting and extrapolation to $\theta = 90°$ were carried out by assuming linear relationship between calculated lattice parameters and the following equation values,

$$\frac{\cos^2\theta}{\sin\theta} + \frac{\cos^2\theta}{\theta}.$$

Correlation length, $L$ (nm), was determined using the following relation,

$$L = \frac{\lambda}{2\Delta\theta\cos\theta}.$$

Here, $\lambda$ (nm) and $\Delta\theta$ (rad) denote wave length of X-rays and peak width (full width at half maxima), respectively. In this study, the following three reflections, 332002, 8 13 5, and 583 reflections, located approximately at $2\theta = 64.8°$, were used for the QC, the 2/1 and 1/1ACs, respectively. To estimate $\Delta\theta$, the peaks were decomposed into two parts originating from Cu-$K\alpha_1$ ($\lambda = 0.15405$ nm) and $K\alpha_2$ by assuming pseudo-Voigt function for each peak shape.

**Physical properties measurements.** The physical properties were measured using one $^3$He refrigerator and four $^3$He/$^4$He dilution refrigerators (each having a different base temperature) installed at Nagoya and Tohoku Universities. Different measurement techniques were taken depending on the temperature region measured: for the electrical resistivity, a four-terminal dc or ac method was taken; for the ac magnetic susceptibility, the mutual inductance method or a SQUID magnetometer; for the heat capacity, the quasi-adiabatic heat-pulse, or relaxation method. The dc magnetization measurement was done using a SQUID magnetometer.

**Data availability.** The data that support the findings of this study are available from the corresponding author (kensho@cc.nagoya-u.ac.jp) upon request.

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

## Acknowledgements

We thank A. Yamamoto for his help of sample preparation and thank Y. Hara for his assistance of low-temperature magnetization measurement. We also thank H. Takakura for his help to create the figure of the QC geometrical structure, and S. Sakai & N. Takemori for useful discussions. This work was financially supported by JSPS KAKENHI (Nos. 26610100, 15H02111, 15H03685, and 16H01071), and Program for Leading Graduate Schools "Integrative Graduate Education and Research Program in Green Natural Sciences", MEXT, Japan.

## Author contributions

K.K. and T.T. carried out the sample preparation. K.K. and T.I. made the sample characterization. K.K., K.I. and K.D. carried out measurements of the electrical resistivity, specific heat, and ac magnetic susceptibility down to about 80 mK. N.K., K.K. and A.O. performed the measurements of heat capacity and the electrical resistivity down to 20–30 mK. N.W. measured the dc and ac magnetization down to about 40 mK. N.K.S. designed the project and drafted the manuscript together with T.I., K.D. and K.I. All authors participated in the writing and review of the final draft.

## Additional information

**Competing interests:** The authors declare no competing financial interests.

