## [Peer Review File · Nature Communications]

Reviewer #1 (Remarks to the Author):

The manuscript presents the observation of superconductivity in an AlZnMg icosahedral quasicrystal. Results are compared to the one obtained for the 2/1 periodic approximant and for a series of 1/1 periodic approximant with different Al content.

All samples display a superconductivity transition, and measurements are consistent with a bulk superconductivity.

The critical temperature is correlated to the inverse $1/\gamma$. The results are also interpreted in the framework of the BCS theory, which reproduces reasonably the results both for the approximant and the quasicrystal. A detailed study of T_c as a function of the upper critical field is also in agreement with the theory of 'dirty' system (although this point should be more elaborated, see further remarks).

The paper and the results are of importance for the solid-state physicist community and present a broad interest.

However the present manuscript should be modified according to the following remarks before publication.

- General reader readability: The introduction and the main part of the text is somehow difficult to read for a non specialist. Effort should be made in particular in the introduction to better introduce the difficult subject of quasicrystals.

The same is true for model for superconductivity. A few sentences should explain what is a 'dirty' systems when presenting the WHH arguments.

- In the introduction or discussion, comparison should be made with other aperiodic crystals for which superconductivity exist.

- Indeed system with incommensurate CDW for instance, do display superconductivity. These are long range ordered systems without periodicity although from a different class of systems.

- The number of references should be increased: for instance which quasicrystal indexing scheme is used to determine the 6D lattice parameter? Which convention is used here? Some more references on amorphous superconducting system (and more recent one) should be added etc...

- Sample characterisation: the 10-fold electron diffraction pattern should be inserted in the main text. There is a lot of free space available, and authors should use it as much as possible. The 2-fold diffraction pattern of both QC and one approximant could be also display: this would nicely illustrate the relation between QC and approximant. Finally a discussion on the quality of the quasicrystalline order should be included in the main text. IS there a linear phason strain? It seems that very few Bragg peaks are visible as compared to other QC.

BCS theory: whereas the BCS theory seems to reproduce the data shown in the main manuscript, it does not seems to do such a good job for the 1/1 approximant when the Al concentration is close to the one of the quasicrystal. This point should be discusses. Is it a consequence of a large disorder?

The hypothesis for the BCS theory should be discussed and at least qualitatively compared to the case of quasicrystal.

WHH theory and 'dirty' systems: i) an explanations of this model, its hypothesis and its applicability should be made in the manuscript. What is a dirty system? How is it characterised? Is a quasicrystal in this class of system?

ii) Why the authors did not plot the entire WHH curve and compared it to the data, since this model calculation is available.

iii) Why this theory is needed, since the BCS theory is used in the previous part of the manuscript. Does it mean there is a breakdown of the BCS theory?

Reviewer #2 (Remarks to the Author):

The authors report on specific heat, magnetization, and electrical resistivity measurements of Al-Zn-Mg quasicrystals and its approximant crystals. They claim that their measurements represent "the discovery of the first superconducting quasicrystal". I do not recommend publication of this manuscript in Nature Communications, mainly for reasons relating to its novelty. I do not believe this is the first report of bulk superconductivity in a quasicrystal, for the following reasons:

1) On page 3 the authors state that: "the superconducting phase of the Al-Zn-Mg alloy was originally considered as QC but later found to be AC", and cite Refs 8 and 12, respectively. Ref 8 studies different alloys than claimed by the authors, namely Al-Cu-Li and Al-Cu-Mg, and reports superconductivity in both of them. Ref 12 does not refute this finding: it does not study the same alloys as Ref 8, but Al-Zn-Mg, does not cite the Ref 8, and does not discuss superconductivity.

2) Just above the previous statement, the authors claim that: "In spite of extensive studies, however, bulk superconductivity is not yet established experimentally in QCs". They cite Ref 10, which reports superconductivity in an Al-Zn-Mg quasicrystal, finding that "superconductivity is indeed a bulk effect". To justify this claim, Ref 10 estimates an "upper limit to any contribution from a volume of normal metal" below the superconducting T_c , finding a value of 3% for the quasicrystal.

Reviewer #3 (Remarks to the Author):

The authors have fabricated high quality QC and AC samples, and used a series of experimental techniques in order to characterize them. The article is well written and the work is explained in detail. An important point in the results is the fact that, the authors have discarded that Al can be responsible for the superconductivity. I suggest that this manuscript can be published in the present form, and of course the authors can also consider my comments and suggestions.

1) In figure 1a, the sample 2/1 AC shows a negative resistance, or is it some mistake with the scaling?

2) In my opinion, the normalization in Fig 1a is not necessary to do at 280K, is enough if you present a normalization at 10K in order to clearly show the superconducting transition.

3) The authors mentioned many times in the main text, also in the supplementary information, that at 15% Al content, the sample shows some anomalies like in the superconducting critical temperature, the electronic specific heat coefficient, the resistivity, however, there is no explanation given. In my opinion, this could be related to electronic stabilization (like Hummel-Rothery phase) as shown in the phase diagram Fig1b. These anomalies were reported for other kinds of QC alloys like AlCuFe [L1], AlPdRe [L2] and some amorphous precursors for the QC phase [L3]. They were attributed to hybridisation of Al *s-p* electrons, which is optimal at such concentrations.

[L1] DOI: 10.1016/S0921-5093(00)01159-X

[L2] DOI: 10.1088/0953-8984/12/47/302

[L3] DOI: 10.1016/j.jnoncrysol.2003.12.001

In the following, we address point by point the questions/comments/concerns of the reviewers. (The comments of the reviewers are written in blue, while our reply to them is written in black.) We hope that the reviewers will agree with our modifications.

Response to Reviewer #1:

The manuscript presents the observation of superconductivity in an AlZnMg icosahedral quasicrystal. Results are compared to the one obtained for the 2/1 periodic approximant and for a series of 1/1 periodic approximant with different Al content.

All samples display a superconductivity transition, and measurements are consistent with a bulk superconductivity.

The critical temperature is correlated to the inverse $1/\gamma$. The results are also interpreted in the framework of the BCS theory, which reproduces reasonably the results both for the approximant and the quasicrystal. A detailed study of T_c as a function of the upper critical field is also in agreement with the theory of 'dirty' system (although this point should be more elaborated, see further remarks).

The paper and the results are of importance for the solid-state physicist community and present a broad interest. However the present manuscript should be modified according to the following remarks before publication.

Reply: First of all, we thank the reviewer for the positive comment and detailed suggestions for improvement of our manuscript. We modified our manuscript following your suggestions.

- General reader readability: The introduction and the main part of the text is somehow difficult to read for a non specialist. Effort should be made in particular in the introduction to better introduce the difficult subject of quasicrystals.

The same is true for model for superconductivity. A few sentences should explain what is a 'dirty' systems when presenting the WHH arguments.

Reply: We added new figures (Fig.1a-c) in the introduction part for non-specialists of quasicrystals. For the explanation of the dirty superconductivity and the WHH theory, we added some sentences in the section of "Superconducting critical field."

- In the introduction or discussion, comparison should be made with other aperiodic crystals for which superconductivity exist.

- Indeed system with incommensurate CDW for instance, do display superconductivity. These are long range ordered systems without periodicity although from a different class of systems.

Reply: We compared the superconductivity between the present QC and density wave superconductors in “Discussion”.

- The number of references should be increased: for instance which quasicrystal indexing scheme is used to determine the 6D lattice parameter? Which convention is used here? Some more references on amorphous superconducting system (and more recent one) should be added etc...

Reply: We added some references concerning the indexing scheme etc. For review of superconductivity in many systems including dirty systems, we added Ref.25 and 27.

- Sample characterisation: the 10-fold electron diffraction pattern should be inserted in the main text. There is a lot of free space available, and authors should use it as much as possible. The 2-fold diffraction pattern of both QC and one approximant could be also display: this would nicely illustrate the relation between QC and approximant. Finally a discussion on the quality of the quasicrystalline order should be included in the main text. IS there a linear phason strain? It seems that very few Bragg peaks are visible as compared to other QC.

Reply: We inserted the 10-fold electron diffraction pattern in the main text. We also added the 2-fold patterns of the 2/1 and 1/1 ACs, which were not included in the original paper. We further added the comments/sentences on the quality of the samples used here in the main text; for example, “the present sample contains a linear phason strain”.

BCS theory: whereas the BCS theory seems to reproduce the data shown in the main manuscript, it does not seems to do such a good job for the 1/1 approximant when the Al concentration is close to the one of the quasicrystal. This point should be discusses. Is it a consequence of a large disorder?

The hypothesis for the BCS theory should be discussed and at least qualitatively compared to the case of quasicrystal.

Reply: We added sentences stating the relation between the sample quality and the physical properties in the last paragraph of “Sample characterization”. The broad nature of the transition is considered as a result of the sample inhomogeneity, as you suggested. We added the sentences in “Discussion” concerning the requirement conditions that superconductivity (the BCS theory) assumes, and we also added the sentences stating the relation to QCs.

WHH theory and 'dirty' systems: i) an explanation of this model, its hypothesis and its applicability should be made in the manuscript. What is a dirty system? How is it characterised? Is a quasicrystal in this class of system?

ii) Why the authors did not plot the entire WHH curve and compared it to the data, since this model calculation is available.

iii) Why this theory is needed, since the BCS theory is used in the previous part of the manuscript. Does it mean there is a breakdown of the BCS theory?

Reply: (i) We added sentences in the text to explain the WHH theory and the dirty systems. QCs should be distinguished from the dirty systems, but the superconductivity of the QC can be described as a dirty superconductor.

(ii) We plotted the entire WHH curve in the revised version.

(iii) BCS theory only describes the zero field state. The WHH theory describes the superconducting properties under magnetic field.

Response to Reviewer #2:

The authors report on specific heat, magnetization, and electrical resistivity measurements of Al-Zn-Mg quasicrystals and its approximant crystals. They claim that their measurements represent "the discovery of the first superconducting quasicrystal". I do not recommend publication of this manuscript in Nature Communications, mainly for reasons relating to its novelty. I do not believe this is the first report of bulk superconductivity in a quasicrystal, for the following reasons:

1) On page 3 the authors state that: "the superconducting phase of the Al-Zn-Mg alloy was originally considered as QC but later found to be AC", and cite Refs 8 and 12, respectively. Ref 8 studies different alloys than claimed by the authors, namely Al-Cu-Li and Al-Cu-Mg, and reports superconductivity in both of them. Ref 12 does not refute this finding: it does not study the same alloys as Ref 8, but Al-Zn-Mg, does not cite the Ref 8, and does not discuss superconductivity.

Reply: We apologize for our wrong numbering in references: Ref.8 (in the first draft) should be corrected to Ref.10. Ref.10 (Graebner & Chen) studied $Mg_3Zn_3Al_2$, and Ref.12 (Takeuchi & Mizutani) also studied the Mg-Zn-Al systems covering $Mg_3Zn_3Al_2$. As seen in FIG.1 of "Takeuchi & Mizutani paper" and also in the inset of Fig.3a in our revised manuscript, $Mg_3Zn_3Al_2$ is close to 1/1AC. Then, in the revised manuscript, we stated as follows:

“there is no QC presenting the convincing evidence for bulk superconductivity¹²⁻¹⁴, i.e., zero resistivity, Meissner effect, heat capacity jump, and the 5-fold rotational

symmetry as well. ... As a test material, we choose the Al-Zn-Mg system owing to two reasons: First, it contains both QC^{15,16} and AC phases¹⁶⁻¹⁸, and second, the AC phase shows superconductivity¹⁴. (In Ref. 14, Mg₃Zn₃Al₂ was considered as QC, but it seems to be AC according to the phase diagram given in Ref. 16 and the present study, see below.)

References

12. Azhazha, V., Grib, A., Khadzhay, G., Malikhin, S., Merisov, B. Pugachov, A. Superconductivity of Ti–Zr–Ni alloys containing quasicrystals, *Phys. Lett. A* **303**, 87-90 (2002).
13. Wagner, J.L., Biggs, B.D., Wong, K.M. & Poon, S.J. Specific-heat and transport properties of alloys exhibiting quasicrystalline and crystalline order. *Phys. Rev. B* **38**, 7436-7441 (1988).
14. Graebner, J.E. & Chen, H.S. Specific heat of an icosahedral superconductor, Mg₃Zn₃Al₂. *Phys. Rev. B* **58**, 1945-1948 (1987).
15. Ramachandrarao, P., & Sastry. G.V.S. A basis for the synthesis of quasicrystals. *Pramana*. **25**, L255-230 (1985).
16. Takeuchi, T. & Mizutani, U. Electronic structure, electron transport properties, and relative stability of icosahedral quasicrystals and their 1/1 and 2/1 approximants in the Al-Mg-Zn alloy system. *Phys. Rev. B* **52**, 9300-9309 (1995).
17. Sugiyama, K., Sun, W. & Hiraga, K. Crystal structure of a cubic Al₁₇Zn₃₇Mg₄₆; a 2/1 rational approximant structure for the Al-Zn-Mg icosahedral phase. *J. Alloys Comp.* **342**, 139-142 (2002).
18. Bergman, G., Waugh, J.L.T., & Pauling, L. The crystal structure of the metallic phase Mg₃₂(Al, Zn)₄₉. *Acta Crystallogr.* **10**, 254-259 (1957).

2) Just above the previous statement, the authors claim that: "In spite of extensive studies, however, bulk superconductivity is not yet established experimentally in QCs". They cite Ref 10, which reports superconductivity in an Al-Zn-Mg quasicrystal, finding that "superconductivity is indeed a bulk effect". To justify this claim, Ref 10 estimates an "upper limit to any contribution from a volume of normal metal" below the superconducting T_c, finding a value of 3% for the quasicrystal.

Reply: We agree that the superconductivity in $\text{Mg}_3\text{Zn}_3\text{Al}_2$ of Ref.10 (Graebner & Chen) is a bulk effect. However, I am very sorry to say that $\text{Mg}_3\text{Zn}_3\text{Al}_2$ is not QC but 1/1AC as mentioned above.

Response to Reviewer #3:

The authors have fabricated high quality QC and AC samples, and used a series of experimental techniques in order to characterize them. The article is well written and the work is explained in detail. An important point in the results is the fact that, the authors have discard that Al can be responsible for the superconductivity. I suggest that this manuscript can be published in the present form, and of course the authors can also consider my comments and suggestions.

Reply: First of all, we thank the reviewer for the recommendation of our manuscript for publication.

1) In figure 1a, the sample 2/1 AC shows a negative resistance, or is it some mistake with the scaling?

Reply: The negative resistivity is ascribed to the problem of ac technique of the resistivity measurement: when the resistivity is very small (i.e., almost zero), the signal is sometimes recorded as a (small) negative voltage.

2) In my opinion, the normalization in Fig 1a is not necessary to do at 280K, is enough if you present a normalization at 10K in order to clearly show the superconducting transition.

Reply: Yes, the normalization at 10 K is OK if we concentrate on the low temperature part. In the present paper, we intended to show the negative temperature coefficient of resistivity, $d\rho/dT < 0$, at high temperatures. This is the reason why the normalization was done at 280 K.

3) The authors mentioned many times in the main text, also in the supplementary information, that at 15% Al content, the sample shows some anomalies like in the superconducting critical temperature, the electronic specific heat coefficient, the resistivity, however, there is no explanation given.

In my opinion, this could be related to electronic stabilization (like Humme-Rothery phase) as shown in the phase diagram Fig1b.

This anomalies were reported for other kinds of QC alloys like AlCuFe [L1], AlPdRe

[L2] and some amorphous precursors for the QC phase [L3]. They were attributed to hybridisation of Al_s-p electrons, which is optimal as such concentrations.

[L1] DOI: 10.1016/S0921-5093(00)01159-X

[L2] DOI: 10.1088/0953-8984/12/47/302

[L3] DOI: 10.1016/j.jnoncrysol.2003.12.001

Reply: We thank you very much for giving us detailed information. The similar argument was done by Takeuchi & Mizutani for the present system, and hence we stated in the text as “Note that γ slightly drops at 15% Al content, which is likely related to the electronic stabilization effect, i.e., the pseudogap formation due to the so-called Hume-Rothery mechanism¹⁶”.

References

16. Takeuchi, T. & Mizutani, U.

Electronic structure, electron transport properties, and relative stability of icosahedral quasicrystals and their 1/1 and 2/1 approximants in the Al-Mg-Zn alloy system.

Phys. Rev. B **52**, 9300-9309 (1995).

Reviewer #1 (Remarks to the Author):

The resubmitted manuscript answers all the remarks of the referees.

The necessary theoretical background is now better introduced; new figures and interpretation have been added (electron diffraction pattern, comparison with the WHH theory).

These important results will certainly attract the attention of a large scientific community in condensed matter physics, strongly correlated systems and quasicrystal in particular.

I strongly recommend the publication of the paper in the present form.

I have a few minor remarks:

- A better care should be taken when identifying periodic crystals versus aperiodic one's: periodic crystals are not 'normal crystals', and QC 'abnormal'. They all are crystals

Example 'while traditional crystals can possess only' should write 'while periodic crystals can possess only'. There are several instances where it should be specified.

- density wave should be changed to charge density wave and abbreviation DW changed to CDW.

- It seems that the reference of the indexing scheme used to index the QC is not given. This should be specified since there are many different schemes, with even different definition of the 6D lattice parameter.

Point-by-point reply to Referee #1

Referee comment: The resubmitted manuscript answers all the remarks of the referees. The necessary theoretical background is now better introduced; new figures and interpretation have been added (electron diffraction pattern, comparison with the WHH theory).

These important results will certainly attract the attention of a large scientific community in condensed matter physics, strongly correlated systems and quasicrystal in particular.

I strongly recommend the publication of the paper in the present form.

We greatly thank the referee for his/her recommendation of our manuscript for publication.

I have a few minor remarks:

- A better care should be taken when identifying periodic crystals versus aperiodic one's: periodic crystals are not 'normal crystals', and QC 'abnormal'. They all are crystals. Example 'while traditional crystals can possess only' should write 'while periodic crystals can possess only'. There are several instances where it should be specified.

Reply:

Following the suggestion, we made changes as follows;

traditional crystal \Rightarrow periodic crystal

Referee comment:

- density wave should be changed to charge density wave and abbreviation DW changed to CDW.

Reply:

Following the suggestion, we made changes as follows;

density wave \Rightarrow charge density wave

DW \Rightarrow CDW

Referee comment:

- It seems that the reference of the indexing scheme used to indexed the QC is not given. This should be specified since there are many different schemes, with even different definition of the 6D lattice parameter.

Reply:

To make clear what we made, we added the following paragraph.

For the QC, the following indexing scheme of the reflection vector \mathbf{g} is used in this paper;

$$\mathbf{g} = \frac{1}{a_{6D}} \sum_{i=1}^6 m_i \mathbf{e}_{i\parallel}.$$

Here, the set of integers, m_i , represents reflection index. The vectors $\mathbf{e}_{i\parallel}$ have a length equal to $1/\sqrt{2}$, and they are parallel to the lines connecting the center of an icosahedron and the surrounding six vertices as in Fig. 6 of Ref. 19. The lattice parameter a_{6D} of the 6D hypercubic lattice may be related to the edge length a_R of the rhombohedral cells of the 3D Penrose tiling as follows,

$$a_R = a_{6D}/\sqrt{2}.$$